# Parasitic Effects on the Congenital Transmission of *Trypanosoma cruzi* in Mother–Newborn Pairs

**DOI:** 10.3390/microorganisms12061243

**Published:** 2024-06-20

**Authors:** Ana Gabriela Herrera Choque, Washington R. Cuna, Simona Gabrielli, Simonetta Mattiucci, Roberto Passera, Celeste Rodriguez

**Affiliations:** 1Unidad de Inmunología Parasitaria, Facultad de Medicina, Universidad Mayor de San Andrés, La Paz, Bolivia; dra_anagherrera@yahoo.com (A.G.H.C.); washingtoncuna@gmail.com (W.R.C.); 2Department of Public Health and Infectious Diseases, Sapienza University of Rome, 00185 Rome, Italy; simona.gabrielli@uniroma1.it (S.G.); simonetta.mattiucci@uniroma1.it (S.M.); 3Department of Medical Sciences, University of Torino, 10126 Torino, Italy; passera.roberto@gmail.com

**Keywords:** *Trypanosoma cruzi*, congenital, placenta, human, parasitemia, DNA, IFN-γ

## Abstract

Maternal parasitemia and placental parasite load were examined in mother–newborn pairs to determine their effect on the congenital transmission of *Trypanosoma cruzi*. Parasitemia was qualitatively assessed in mothers and newborns by the microhematocrit test; parasite load was determined in the placental tissues of transmitting and non-transmitting mothers by the detection of *T. cruzi* DNA and by histology. Compared to transmitter mothers, the frequency and prevalence of parasitemia were found to be increased in non-transmitter mothers; however, the frequency and prevalence of parasite load were higher among the transmitter mothers than among their non-transmitter counterparts. Additionally, serum levels of interferon (IFN)-γ were measured by an enzyme-linked immunosorbent assay (ELISA) in peripheral, placental, and cord blood samples. Median values of IFN-γ were significantly increased in the cord blood of uninfected newborns. The median IFN-γ values of transmitter and non-transmitter mothers were not significantly different; however, non-transmitter mothers had the highest total IFN-γ production among the group of mothers. Collectively, the results of this study suggest that the anti-*T. cruzi* immune response occurring in the placenta and cord is under the influence of the cytokines from the mother’s blood and results in the control of parasitemia in uninfected newborns.

## 1. Introduction

The etiological agent of Chagas disease, *Trypanosoma cruzi*, is responsible for the endemicity of this disease in the Americas, together with risk factors strongly linked to socioeconomic factors, such as poverty and marginalization. The parasite is transmitted most often by blood-sucking triatomine bugs, but also through blood transfusion, organ transplant, congenital transmission, or oral transmission in the sylvatic cycle. Although control programs have made possible the reduction of the prevalence of *T. cruzi* infection [1,2], uncontrolled transmission from mothers infected with *T. cruzi* to their fetuses is a potential threat to spreading this infection over time, which can be recurrent at each pregnancy and transmitted from one generation to the next [3]. The incidence of *T. cruzi* infection in pregnant women varies from 6% to 54% [4,5,6,7]. Transmission from mothers chronically infected with *T. cruzi* to newborns, occurring in 1–10% of pregnancies [8,9,10], can be associated with severe disease and mortality [11,12]. 

Despite many experimental approaches, there is still no clear understanding of the factors associated with congenital Chagas disease. Among these factors, high maternal parasitemia and diminished IFN-γ production [8,13,14,15,16] have been postulated to account for transplacental transmission, but none of these issues has been unequivocally implicated as crucial in the transmission process.

Given the crucial role of IFN-γ in the context of this study, its concentration was measured in maternal, placenta, and cord blood samples of uninfected and infected mothers, and their newborns. In particular, IFN-γ production by whole blood cells upon specific stimulation observed in pregnant mothers limits the occurrence of newborn congenital infection [17]; depressed production of IFN-γ by blood cells of pregnant women was associated with congenital transmission of *T. cruzi* [8]; a study by Hermann and collaborators revealed a reduced production of IFN-γ by cord blood NK cells obtained from congenitally infected newborns [18]; and in our previous study, increased production of IFN-γ elicited by enhanced parasitemia was observed in maternal, placenta, and cord samples of chronically infected mothers delivering uninfected newborns [19]. These results point to a strong pro-inflammatory immune response in the absence of congenital transmission.

Within the scope of this study, the median and total median values from the mother, placenta, and cord were compared between samples from mothers of uninfected newborns (non-transmitters) and mothers of infected newborns (transmitters). Moreover, parasite load at the tissue level was measured through molecular analysis and microscopic observation of placental tissues from pregnant women chronically infected with *T. cruzi*. The ensuing question is to define which factors, such as parasitemia, parasitic load, and cytokine interactions, in peripheral, placental, and cord blood are involved in the outcome of parasite transmission to newborns.

## 2. Materials and Methods

### 2.1. Study Population and Sample Collection

The study was carried out at the maternity hospital Odón Ortega in Yacuiba, South Bolivia, an endemic area for *T. cruzi* infection. Informed consent for participation was obtained from women (14 transmitters, 15 non-transmitters) before their inclusion in the study. Maternal information (i.e., age, village of birth and village of residence, medical history, and present symptoms) was documented at enrollment. Women with concomitant infections (tuberculosis, toxoplasmosis) were excluded from the study. Peripheral blood samples were drawn by venipuncture from all pregnant women before delivery, and cord blood was collected after the umbilical section. Placental blood was obtained from the maternal side by rinsing several times with a sterile physiological solution (0.9% NaCl), as previously described [20]. Briefly, the excision of a 5-cm^3^ block of tissue from the cleaned maternal side permitted aspiration of intervillous blood using a sterile Pasteur pipette. Thereafter, blood samples were centrifuged to separate the packed erythrocytes, and the serum was frozen at −80 °C until assayed for antibodies. *Trypanosoma cruzi* infection in pregnant women was determined by specific serological tests, that is, indirect hemagglutination (HAI, Chagas, Polychaco S.A.I.C., Buenos Aires, Argentina), with a sensitivity and specificity of 99%, followed by enzyme-linked immunosorbent assays (first- and second-generation, Wiener Laboratories, Buenos Aires, Argentina), with sensitivity and specificity between 98–99%, for confirmation of diagnosis. The maternal parasitemia in chronic chagasic mothers and infection in neonates were qualitatively assessed by detecting living parasites through microscopic examination of the buffy coat from the mother’s peripheral and cord blood by the microhematocrit method because of its ease of performance, reliability, and high sensitivity (97.4%) in infants under 6 months old [21]. Blood was collected in four 75-μL microhematocrit tubes, for a volume of 300 µL in total, and the results were scored as “detectable” and “undetectable”, denoting higher and lower parasitemia, respectively. Newborns’ negative microhematocrit test results were confirmed at one month of age in case the peak of parasitemia was observed [9,22,23]. Hereinafter, detectable and undetectable parasitemia will be referred to as positive microhematocrit (+µHT) and −µHT, respectively. Samples from non-infected mothers (controls) were defined as samples from mothers with negative serology for *T. cruzi* and negative µHT test results in which there was no DNA amplification. The study received approval from the ethics committee of the Faculty of Medicine and the Medical College of Bolivia (Appendix A).

### 2.2. Tissues 

Placental tissue sections 1 cm long and adjacent to the umbilical cord insertion were obtained from each parturient woman immediately after delivery and washed in sterile physiologic saline solution. One tissue sample was kept frozen at −80 °C until molecular analysis, while another one was prepared for tissue staining.

### 2.3. Paraffin-Embedded Tissue Sections

Tissue sections were fixed in buffered formalin (3.7% formaldehyde in 10 mM phosphate buffer, pH 7.4), dehydrated through immersion in an alcohol series (70–100%) for 20 min each, and then ‘cleared’ by placing the tissues into a xylene bath for 20 min. Next, the tissues were embedded in molten paraffin in a mold. Sections (3–4 μm) were cut and affixed onto glass slides coated with aminoalkylsilane (Sigma-Aldrich, St. Louis, MO, USA), designed to enhance tissue adhesion. 

### 2.4. Hematoxylin and Eosin Staining and Bright-Field Microscopy

Slides were stained with hematoxylin and eosin (H&E) solution (~30 s), thoroughly rinsed with distilled water then dehydrated to xylene and permanently mounted. Stained samples were examined microscopically using an Olympus BX53 microscope (Shinjuku-ku, Tokyo, Japan), provided with an Olympus DP73 high-performance Peltier-cooled digital camera (Olympus Corporation, Shinjuku-ku, Tokyo, Japan).

### 2.5. Molecular Analysis

DNA was extracted from placental tissue samples using a commercial kit (DNeasy Blood & Tissue Kit, Qiagen, GmbH, Hilden, Germany) according to the manufacturer’s instructions. Extracted DNAs were analyzed using the NanoDrop 2000 spectrophotometer (Thermo Scientific, Middlesex, MA, USA) and electrophoresed on 1% agarose gel to determine the quantity and quality of the extracted DNA as a means to verify that DNA was being extracted correctly. Each DNA was submitted to a TaqMan Real-time (RT)-PCR assay to amplify a region of the 18S ribosomal RNA (rRNA) gene of *T. cruzi* (Genesig Primer- Design, Camberley, UK) according to the manufacturer’s instructions. In addition, RT-PCR positive samples were subjected to a nested PCR (N-PCR) to perform a sequence analysis of amplified products. Briefly, nuclear DNA was first amplified using primers TCZ1 (5′-CGAGCTCTTGCCACACGGG-3′) and TCZ2 (5′-CCTCCAAGCAGCGGATAGTTCAGG-3′), to yield 188 base pairs (bp) product using the described protocol, followed by the N-PCR reaction with TCZ3 (5′-TGCTGCASTCGGCTGATCGTTTTCGA-3′) and TCZ4 (5′-CAR GSTTGTTTGGTGTCCAGTGTTGTGA-3′), which yield a product of 149 bp for sequence analysis [24]. The first PCR amplification was performed in 25 μL volumes under the following final conditions: 1× buffer including 1.5 mM MgCl_2_, 0.2 mM of each deoxynucleoside triphosphate (dNTP), 1 μM each of forward and reverse primers, and 1 U of *Taq* polymerase (BIOTAQTM DNA Polymerase, Aurogene, Rome, Italy). The thermal profile used was as follows: 94 °C for 30 s, 60 °C for 30 s, and 72 °C for 1 min for 30 cycles, followed by a final extension for 7 min at 72 °C. One microliter of the reaction was used for the second amplification, in which primers TCZ3 and TCZ4 amplified a 149-nucleotide internal sequence of the same repetitive sequence. The N-PCR conditions and protocol were the same as for the first amplification. Amplicons were purified (SureClean Bioline, Aurogene) and then sequenced (Eurofins MWG Operon, Ebersberg, Germany). Sequences corrected by visual analysis of the electropherograms and aligned using ClustalW (http://www.genome.jp/tools/clustalw (accessed on 10 November 2018), were compared with those available in the GenBank (http://www.ncbi.nlm.nih.gov/genbank/ (accessed on 10 November 2018) dataset by BLAST analysis.

### 2.6. Measurement of IFN-γ in Peripheral, Placental, and Cord Serum

Peripheral, placental, and cord blood derived-serum samples were analyzed by sandwich ELISA, using pairs of capture and biotinylated-specific secondary antibodies (BioSource Europe S.A., Nivelles, Belgium). A standard curve of recombinant human cytokine was run with each plate. The sensitivity of the IFN-γ assay is 0.03 IU/mL. For a more representative comparison of the cytokine response between uninfected, transmitter, and non-transmitter mothers, the total production of IFN-γ from the three sites—periphery, placenta, and cord—was compared between the groups of mothers.

### 2.7. Statistical Analysis

Continuous covariates were described as the median (IQR, or interquartile range), while categorical ones were described as absolute/relative frequencies. A non-parametric method for inferential tests on independent data was employed. The differences between continuous variables were assessed by the Kruskal–Wallis test, using the Mann–Whitney test for post hoc analysis. All reported *p*-values were obtained by the two-sided exact method at the conventional 5% significance level. Data were analyzed as of May 2024 using R 4.4.0 (R Foundation for Statistical Computing, Vienna, Austria).

## 3. Results

### 3.1. Characterization of T. cruzi-Infected Mothers and Newborns

The frequency (i.e., number of times) of maternal parasitemia as well as the parasite load of *T. cruzi*-infected pregnant women was determined by µHT test and RT-PCR, respectively. A higher frequency of patent parasitemia in non-transmitters (12/15) than in transmitter mothers (6/14) was associated with a lower frequency of parasite load in non-transmitters (5/15) than transmitter mothers (14/16). Congenital transmission determined by µHT findings was observed in 15 newborns of 14 infected mothers. Of note, a transmitter mother (Code 1072) had triplet newborns, two infected and one uninfected. (Table 1).

### 3.2. RT-PCR Assay

Molecular analyses of tissue samples from the placenta of transmitter and non-transmitter mothers were positive for *T. cruzi* DNA in 87.5% (14/16) and 33.3% (5/15) of cases, respectively. Positivity amount of the target DNA ranged from 10^1^ to 10^4^ copies/μL. Two placental samples of transmitter mothers with +μHT test results (2097, 1961) were RT-PCR negative (Table 1 and Table 2). All positive samples on RT-PCR assay were also confirmed with the N-PCR assay (Figure 1) and the amplicons of 149 base pairs were correctly sequenced. The sequences obtained showed high nucleotide identity (98–100%) with those from *T. cruzi* repetitive DNA sequences available in the GenBank database (accession number KX235537).

### 3.3. Association between IFN-γ, Parasitic Factors, and Newborn Infection Status 

As can be seen in Table 2, the pregnant women who did not transmit the parasites to their newborns presented higher IFN-γ production than their transmitter counterparts. Likewise, parasitemia was proportionally higher (i.e., more prevalent) in the non-transmitter (80%) than in the transmitter mothers (42.8%), and this was associated with a reduced prevalence of parasite load (33.3%) compared to transmitter women (87.5%).

### 3.4. Production of IFN-γ in Uninfected, Transmitter, and Non-Transmitter Mothers

Median concentrations of IFN-γ were not significantly different between samples from transmitter and non-transmitter mothers (*p* = 0.224). However, total IFN-γ production was highest in non-transmitter mothers (*p* = 0.002), and uninfected newborns displayed a significantly higher level of IFN-γ than newborns of uninfected or transmitter mothers (*p =* 0.037) (Table 3).

### 3.5. Histological Studies

Histological preparations of placental tissue sections representative of non-transmitter and transmitter mothers are shown in Figure 2 and Figure 3. Different forms of the parasite were detected through the microscopic observation of chorionic villous human placenta from these mothers. Amastigote nests were observed in preparations from non-transmitter (Figure 2A,B) and transmitter (Figure 3A) mothers, while released parasites were found in placentas from non-transmitter (Figure 2C) and transmitter (Figure 3B,C) mothers. 

## 4. Discussion

The main feature of this study was the evaluation of parasitic factors and IFN-γ production in chronically infected pregnant women associated with the occurrence or absence of congenital *T. cruzi* infection. We were able to show an inverse correlation between maternal parasitemia and placental parasite load in the two groups of pregnant mothers. A higher frequency of detectable parasitemia was observed in non-transmitter mothers than in transmitter mothers, but the contrary was observed in transmitter mothers, suggesting that the higher parasitemia was not a transmission risk in our study. This negative correlation was ascribed to higher levels of circulating IFN-γ in non-transmitter than transmitter mothers. It is well known that IFN-γ confers protection from *T. cruzi* infection [25,26,27] through macrophage nitric oxide production to kill parasites in synergy with TNF-α [28,29].

Protection from vertical transmission likely requires the coordination of different components of the immune system. Controlling the parasite load in non-transmitter mothers through the induction of IFN-γ tempts us to speculate that maternal IFN-γ influences the immune response at the level of the placenta and cord and protects the newborn from infection in utero. Relevant to that point, previous studies reported higher levels of cytokines, including IFN-γ, in the periphery and placenta of malaria uninfected compared to infected women, suggesting that these cytokines are involved in the control of parasitemia in peripheral blood and the placenta [30]. Along the same line of reasoning, it was determined that the total IFN-γ concentration was highest in non-transmitter mothers, suggesting a synergistic effect of IFN-γ between the mother, placenta, and cord in the control of congenital infection. In addition, the increased level of this cytokine in the cord of uninfected newborns compared to newborns of uninfected and transmitter mothers probably results from the combined effect of maternal and cord IFN-γ contributing to the prevention of congenital infection. Taking into account the protective role of the placenta, IFN-γ might be transported from the placenta to the cord. Evidence suggests that IFN-γ involved in maternal immune activation can cross the placenta and predispose to neuropsychiatric disorders [31]. Another study points to the induction of the mother’s immune system to release proinflammatory cytokines after infection during pregnancy which can cross the placenta and enter the fetal circulation [32].

Part of the focus of this study was the evaluation of parasite load in the congenital transmission process, through the histological and molecular analyses of placental tissue samples. Patterns of amastigote nests and free parasites were observed in histological samples of infected mothers. However, as determined by RT-PCR analyses suggesting live parasites in the placenta, samples with a higher and lower prevalence of parasite load corresponded with infected and non-infected newborns, respectively. We observed that the interplay between the level of parasitemia and the production of IFN-γ determines the occurrence or absence of congenital infection. Therefore, when parasitemia diminishes the stimulus for the production of this cytokine diminishes, there is less control of parasite load, and congenital infection occurs. Conversely, when parasitemia rises, production of IFN-γ increases, and parasite load and congenital infection are controlled. 

Regarding the placenta, the case of triplets born to a *T. cruzi*-infected mother (Table 1, sample 1072) from a triamniotic trichorionic delivery of which two newborns were infected and one uninfected is indicative of the protection offered by the placenta. Previous reports on the placenta´s role in protecting the fetus from infection point to the effect of lysosomal subfractions of the placenta on parasite viability and infectivity [33], a decreased viability of parasites in villous explants coincident with high levels of nitric oxide production by placental cells [34], and our previous study in chronically infected mothers showing that higher levels of IFN-γ in their placentas prevented *T. cruzi* infection of neonates [19]. Interestingly, placental samples 2097 and 1961 despite being RT-PCR negatives were associated with vertical transmission of *T. cruzi,* suggesting low levels of parasite load in these samples, undetected in the RT-PCR assay. 

Our results demonstrate a potential relation between maternal immune profile during pregnancy and vertical *T. cruzi* transmission. However, more in-depth laboratory testing to identify likely mechanisms of congenital transmission, such as the study of genes differentially expressed in the placenta of transmitter and non-transmitter mothers related to the immune system as well as the polymorphism of some genes expressed in the placenta in association with congenital *T. cruzi* transmission [35,36,37], which was not possible to undertake, could have contributed to improving understanding to our observations. 

A limited number of cases due to budget limitations hindered our ability to have stronger statistical support. Budget support was secured for two years and due to the low incidence of congenital transmission access to these cases is limited. 

## Figures and Tables

**Figure 1 microorganisms-12-01243-f001:**
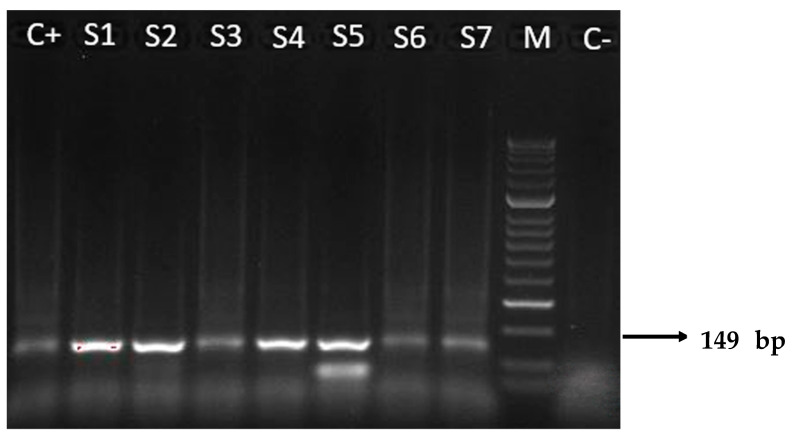
Results of N-PCR amplifications with *T. cruzi* DNA nuclear, electrophoresed on a 2% agarose gel and visualized by ethidium bromide staining. The 149 base pairs (bp) were amplified through N-PCR with primers TCZ3 and TCZ4. M: molecular weight marker (50 bp); C+: positive control (*T. cruzi* II of Y strain); S1–S7: representative amplicons of positive patients, from transmitter (S2–S4) and non-transmitter (S1, S5–S7) mothers; C−: negative control from a patient with negative serology for *T. cruzi*.

**Figure 2 microorganisms-12-01243-f002:**
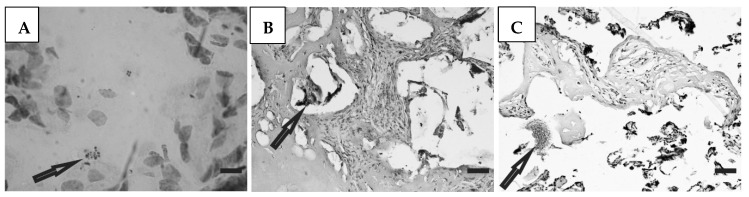
Histological section of the placenta from a non-transmitter mother. Arrows point to (**A**,**B**) amastigote nests; and (**C**) released parasites (H&E). Scale bar: 25 μm.

**Figure 3 microorganisms-12-01243-f003:**
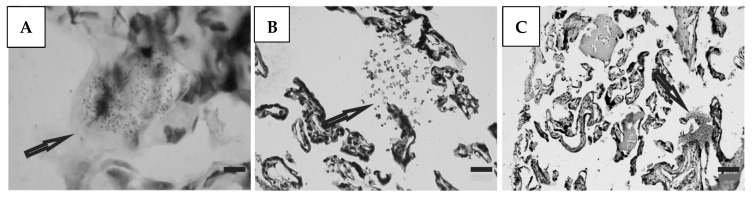
Histological section of the placenta from a transmitter mother. Arrows point to (**A**) amastigote nest; and (**B**,**C**) released parasites (H&E). Scale bar: 25 μm.

**Table 1 microorganisms-12-01243-t001:** Characteristics of mother/newborn pairs.

	Transmitter Mother		Newborn *	Placenta
Code	Serology	µHT	µHT	RT-PCR 18S rRNA
525	+	+	+	+
1072	+	+	+ (a)	+
		+	+ (b)	+
			− (c)	+
1226	+	+	+	+
2016	+	+	+	+
2097	+	+	+	−
1961	+	+	+	−
1056	+	−	+	+
1088	+	−	+	+
1089	+	−	+	+
1211	+	−	+	+
1229	+	−	+	+
1234	+	−	+	+
1977	+	−	+	+
2056	+	−	+	+
	**Non-Transmitter Mother**		**Newborn**	**Placenta**
**Code**	**Serology**	**µHT**	**µHT**	**RT-PCR 18S rRNA**
368	+	+	−	−
1968	+	+	−	−
322	+	+	−	−
323	+	+	−	−
527	+	+	−	−
298	+	+	−	−
461	+	+	−	−
496	+	+	−	−
2115	+	+	−	−
338	+	+	−	+
437	+	+	−	+
1246	+	+	−	+
1150	+	−	−	+
1157	+	−	−	+
1137	+	−	−	−
	**Uninfected Mother**		**Newborn**	**Placenta**
**Code**	**Serology**	**µHT**	**µHT**	**RT-PCR 18S rRNA**
1154	−	−	−	−
1159	−	−	−	−
1161	−	−	−	−
1151	−	−	−	−

* a, b, and c under “Newborn” indicate the triplets born to mother 1072.

**Table 2 microorganisms-12-01243-t002:** Maternal IFN-γ production, parasitemia, parasite load, and newborn parasitemia in transmitter and non-transmitter mothers.

Mother	IFN-γ †	Blood	Placenta	Newborn
		µHT %	RT-PCR %	µHT %
Transmitters	0.59 (0.29–1.13)	42.8 (6/14)	87.5 (14/16) ^a^	93.7 (15/16) ^b^
Non-transmitters	1.30 (0.56–2.65)	80.0 (12/15)	33.3 (5/15)	0.0 (0/15)

† IU/mL, median (interquartile range). ^a, b^, data include triplets born to one mother.

**Table 3 microorganisms-12-01243-t003:** Serum concentrations of IFN-γ from peripheral, placental, and cord blood samples in uninfected, transmitter, and non-transmitter mothers.

	IFN-γ †					
Mother	Periphery	Placenta	Cord	*p* *	Total	*p* *
Uninfected	0.81 (0.37–1.26)	0.24 (0.04–0.86)	−0.08 (−0.17–0.53)		0.33 (−0.3–1.09)	
Transmitter	0.59 (0.29–1.13)	0.65 (0.27–2.37)	0.62 (0.24–1.12)	0.224 ^a^	0.62 (0.24–1.35)	
Non-transmitter	1.30 (0.56–2.65)	1.19 (0.22–1.88)	1.56 (0.12–2.18)	0.037 ^b^	1.39 (0.22–2.47)	0.002 ^c^

† IU/mL, median (interquartile range). * *p* value. ^a^ Comparison between transmitter and non-transmitter mothers. ^b^ Comparisons of values between newborns of uninfected, transmitter, and non-transmitter mothers. ^c^ Comparison of total values between uninfected, transmitter, and non-transmitter mothers.

## Data Availability

The data that support the findings of this study are included within the article.

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
