# Peer review of "Parasitic Effects on the Congenital Transmission of Trypanosoma cruzi in Mother–Newborn Pairs"

_microorganisms, 2024, doi:10.3390/microorganisms12061243_

Round 1

Reviewer 1 Report

Comments and Suggestions for Authors

The manuscript “Parasitic Effects on the Congenital Transmission of Trypanosoma cruzi in Mother-Newborn Binomials“ is an interproposed for the study of the factors such as parasitemia, parasitic load, and cytokine interactions in peripheral, placental, and cord blood that are involved in the parasite transmission to newborns.

Introduction 

The introduction is concise. However, I consider it should include more information about the IFN y, the cells that produce it, and how it could be triggered. If it is modified during pregnancy. Circulatory and placental INF are produced by the same cells, their level and other factors that can affect its expression.  

Materials and methods:

The methods are accurately and clearly described. But didnt you realize the serology test on the newborns? Maybe IgM determination or increasing IgG titles?

Results

Results are correctly described and are consistent with methods. Nevertheless the way tables are presented is not as clear as it could be. I strongly suggest modifying table 2 and 3 to present them as graphics (maybe as box and mustache plot). I consider it would help visualize the results and make them easier to understand for readers.

Discussion

The points discussed are ok, nevertheless, the authors must mention the limitations of the protocol.  They have observed negative correlations among parasitemia and circulating IFN/ parasite transmission, nevertheless, it does not mean that these are the unique variables that interfere in the congenital infection process, so it must be mentioned and discussed in detail mentioning other factors that could interfere.

I consider it to be an interesting work that presents some factors that are involved in the parasite transmission to newborns; However, it is necessary to clarify in discussion the limitations that were presented in the project and whether these could represent a bias in the information presented. 

Author Response

I am grateful to the reviewers for their insightful comments on my paper. I have 
been able to incorporate changes to reflect most of the suggestions provided by 
the reviewers.
Here is a point-by-point response to the reviewers’ comments and concerns.
Comments from Reviewer 1
INTRODUCTION
Comment: 
The introduction is concise. However, I consider it should include more 
information about the IFN y, the cells that produce it, and how it could be 
triggered. If it is modified during pregnancy. Circulatory and placental INF are 
produced by the same cells, their level and other factors that can affect its 
expression.
Response:
In response to the comments above, the role of IFN-γ has been expanded on
the Introduction page 2, lines 49-57.
MATERIALS AND METHODS
Comment: 
The methods are accurately and clearly described. But didnt you realize the 
serology test on the newborns? Maybe IgM determination or increasing IgG 
titles?
Response: 
The observation of cord blood parasites by the microhematocrit test is the 
method of choice for T. cruzi congenital transmission. Though IgM is usually 
interpreted as an indicator of acute infection, the detection of specific IgM 
antibodies in congenital transmission can lead to false positive reactions and 
also to a significant frequency of false negative reactions, due to low 
concentration of IgM antibodies in recent congenital infections, immune 
complex formation between IgM and an excess of antigen, and maternal IgG 
transferred to the fetus that suppresses fetal IgM synthesis. 
RESULTS
Comment: 
Results are correctly described and are consistent with methods. Nevertheless 
the way tables are presented is not as clear as it could be. I strongly suggest 
modifying table 2 and 3 to present them as graphics (maybe as box and 
mustache plot). I consider it would help visualize the results and make them 
easier to understand for readers.
Response:
Thank you for this suggestion. However, I respectfully prefer not to present 
tables 2 and 3 as graphics. After doing a box plot I can not see a substantial 
difference to visualize the results, Tables 2 and 3 are simple where the results 
conveyed are easily grasped.
DISCUSSION
Comment: 
The points discussed are ok, nevertheless, the authors must mention the 
limitations of the protocol. They have observed negative correlations among 
parasitemia and circulating IFN/ parasite transmission, nevertheless, it does not 
mean that these are the unique variables that interfere in the congenital 
infection process, so it must be mentioned and discussed in detail mentioning 
other factors that could interfere.
Response:
I agree that it would have been interesting to expand our research to the 
analysis of other factors related to congenital transmission, such as the study of 
genes differentially expressed in the placenta of transmitter and non-transmitter 
mothers related to the immune system as well as the polymorphism of some 
genes expressed in the placenta in association with T. cruzi congenital 
transmission, mentioned on page 8, lines 280-285 of the Discussion. However, 
the lack of related data on these factors in this study prevented further 
discussion. 
Comment:
I consider it to be an interesting work that presents some factors that are 
involved in the parasite transmission to newborns; However, it is necessary to 
clarify in discussion the limitations that were presented in the project and 
whether these could represent a bias in the information presented.
Response:
The only bias found was the limited number of samples included due to budget 
constraints that impinge upon our ability to have stronger statistical support, 
mentioned on page 8, lines 286-287.

Reviewer 2 Report

Comments and Suggestions for Authors

Line 1. The authors could consider amending the title to stress the main findings of this study, for example ‘Potential protective effect of interferon gamma in congenital transmission of Trypanosoma cruzi’.

Line 16/17. In the abstract, the authors say ‘the frequency and prevalence of..’, but in the main text the difference between ‘frequency’ and ‘prevalence’ is not clearly specified. Perhaps the words ‘the frequency and prevalence of’ can be removed without losing the meaning?

Line 30. Delete ‘mainly’.

Line 54. The authors should clarify at the beginning of this paragraph that it describes the work in the current study.

Line 64. Is Yacuiba locality an area of active transmission by the vector?

Line 66. The authors should state the numbers of individual women in this study (15 non-transmitters + 14 transmitters).

Line 116. The authors should clarify that ‘RT-PCR’ means ‘Real-time PCR’, not ‘Reverse transcriptase PCR’

Line 116-133. The authors should amend the description of the PCRs, to clarify that the first PCR was with primers TCZ1 & TCZ2 to yield a 188bp product using the described protocol. Then, the nested PCR using TCZ3 & TCZ4 was performed on positive samples to generate a 149bp product for sequence analysis.

Line 157. Add the word ‘respectively’ after ‘RT-PCR’

Line 168/171/178/259. For consistency, it should be Tables/Figures 1, 2, etc in the text, not I, II, etc.

Line 168-170. The statement about control mothers would be better placed in the Methods section.

Line 180-181. The data for parasitaemia are already presented in Line 158, so the authors should present the % there.  

Line 181-182. The % for load are already presented in Line 165, so the authors do not need to repeat them.

Table 2. The authors should add a footnote for the transmitters row (placenta and newborn columns) stating that these data include triplets born to one mother.

Line 203. Should the footnote be: ‘c. Comparison of total values between NEWBORNS OF uninfected, transmitter, and non-transmitter mothers’, to match Lines 197-198?

Lines 224/227. Please state scale bar measurement for Figures 1 and 2 (example: scale bar = 1 micron).

Line 214-218. Suggested amendment: ‘We were able to show an inverse correlation between  maternal parasitemia and placental parasite load in THE TWO GROUPS OF pregnant mothers. A higher frequency of detectable parasitemia was observed in non-transmitters compared to transmitter mothers, BUT the contrary was observed in transmitter mothers, suggesting that the higher parasitemia was not a transmission risk in our study’.

Lines 277-278. Perhaps this sentence should be moved to Line 270, to be between the current first and second sentences of that paragraph.

Comments on the Quality of English Language

English language is generally fine.

Author Response

I am grateful to the reviewers for their insightful comments on my paper. I have 
been able to incorporate changes to reflect most of the suggestions provided by 
the reviewers.
Here is a point-by-point response to the reviewers’ comments and concerns.
Comments from Reviewer 2
Comment: 
Line 1. The authors could consider amending the title to stress the main findings 
of this study, for example ‘Potential protective effect of interferon gamma in 
congenital transmission of Trypanosoma cruzi’.
Response: 
Thanks for the suggestion, however, I rather prefer to keep the same title.
Comment:
Line 16/17. In the abstract, the authors say ‘the frequency and prevalence of..’, 
but in the main text the difference between ‘frequency’ and ‘prevalence’ is not 
clearly specified. Perhaps the words ‘the frequency and prevalence of’ can be 
removed without losing the meaning?
Response:
Frequency and prevalence are specified in Results by mentioning: “(i.e., 
number of times)” (page 4, line 164), and “(i.e., more prevalent)” (page 5 line 
187), respectively.
Comment:
Line 30. Delete ‘mainly’.
Reponse:
It has been deleted.
Comment:
Line 54. The authors should clarify at the beginning of this paragraph that it 
describes the work in the current study.
Response:
This has been clarified at the beginning of the paragraph by adding: “Within the 
scope of this study, ……”, on page 2, line 58.
Comment:
Line 64. Is Yacuiba locality an area of active transmission by the vector?
Response:
Yes.
Comment:
Line 66. The authors should state the numbers of individual women in this study 
(15 non-transmitters + 14 transmitters).
Response:
The number of transmitters and non-transmitters mothers are specified on page 
2, line 70.
Comment:
Line 116. The authors should clarify that ‘RT-PCR’ means ‘Real-time PCR’, not 
‘Reverse transcriptase PCR’
Response:
This is specified on page 3, line 123.
Comment:
Line 116-133. The authors should amend the description of the PCRs, to clarify 
that the first PCR was with primers TCZ1 & TCZ2 to yield a 188bp product 
using the described protocol. Then, the nested PCR using TCZ3 & TCZ4 was 
performed on positive samples to generate a 149bp product for sequence 
analysis.
Response:
Accordingly, the description of the PCR has been detailed further as suggested, 
on page 3, lines 127-132.
Comment:
Line 157. Add the word ‘respectively’ after ‘RT-PCR’
Response:
Thanks, this has been corrected on page 4, line 165.
Comment:
Line 168/171/178/259. For consistency, it should be Tables/Figures 1, 2, etc in 
the text, not I, II, etc.
Response:
This has been corrected.
Comment:
Line 168-170. The statement about control mothers would be better placed in 
the Methods section.
Response:
This statement was placed on Materials and Methods, page 2, lines 95-97.
Comment:
Line 180-181. The data for parasitaemia are already presented in Line 158, so 
the authors should present the % there.
Response:
Under “3.1.Characterization of T. cruzi-infected mothers and newborns”, we 
refer to frequency, and in “3.3. Association between IFN-γ, parasitic factors, and 
newborn infection status” we talk about prevalence, so, these were not 
changed.
Comment:
Line 181-182. The % for load are already presented in Line 165, so the authors 
do not need to repeat them.
Response:
Same as above.
Comment:
Table 2. The authors should add a footnote for the transmitters row (placenta 
and newborn columns) stating that these data include triplets born to one 
mother.
Response:
Footnote included in Table 2.
Comment:
Line 203. Should the footnote be: ‘c. Comparison of total values between 
NEWBORNS OF uninfected, transmitter, and non-transmitter mothers’, to 
match Lines 197-198?
Response:
Thanks. It is correct “Comparison of values between newborns…”. It is 
corrected in the footnote. 
Comment:
Lines 224/227. Please state scale bar measurement for Figures 1 and 2 
(example: scale bar = 1 micron).
Response:
A scale bar of 25m is specified in figure footnotes.
Comment:
Line 214-218. Suggested amendment: ‘We were able to show an inverse 
correlation between maternal parasitemia and placental parasite load in THE 
TWO GROUPS OF pregnant mothers. A higher frequency of detectable 
parasitemia was observed in non-transmitters compared to transmitter mothers, 
BUT the contrary was observed in transmitter mothers, suggesting that the 
higher parasitemia was not a transmission risk in our study’.
Response:
Corrections were made on pge 7, lines 222-226.
Comment:
Lines 277-278. Perhaps this sentence should be moved to Line 270, to be 
between the current first and second sentences of that paragraph.
Response:
No changes were made.